# Hexagonal Boron Nitride/Microfibril Cellulose/Poly(vinyl alcohol) Ternary Composite Film with Thermal Conductivity and Flexibility

**DOI:** 10.3390/ma12010104

**Published:** 2018-12-30

**Authors:** Xin Ge, Wei-Jie Liang, Jian-Fang Ge, Xun-Jun Chen, Jian-Ye Ji, Xiao-Yan Pang, Ming He, Xiao-Meng Feng

**Affiliations:** 1College of Chemistry and Chemical Engineering, Zhongkai University of Agriculture and Engineering, Guangzhou 510230, China; 18011779939@163.com (X.G.); cxj.qiao@163.com (X.-J.C.); jjyjasonky@163.com (J.-Y.J.); shelly_pxy@163.com (X.-Y.P.); heming1026@163.com (M.H.); 7y_only@163.com (X.-M.F.); 2School of Materials Science and Engineering, Northwestern Polytechnical University, Xi’an 710072, China; leungvijer@163.com

**Keywords:** hexagonal boron nitride, microfibril cellulose, composite, thermal conductivity, flexibility

## Abstract

Microfibril cellulose (MFC), which is detrimental to soil cultivation and environmental protection, is derived from waste pineapple leaves. Hexagonal boron nitride (h-BN) was modified with polydopamine (PDA)—PDA@h-BN named pBN, and then combined with MFC to prepare a novel hybrid powder. The effect of PDA on h-BN and the binding effect between pBN and MFC were characterized by X-ray photoelectron spectroscopy (XPS), Thermogravimetric (TG), scanning electron microscopy (SEM), and Fourier Transform-Infrared (FT-IR). Poly (vinyl alcohol) (PVA) was used as an eco-friendly polymeric matrix to prepare a pBN-MFC-PVA composite film. The mechanical strength, hydrophobicity, and thermal conductivity of the film were studied and the results confirmed that h-BN was chemically modified with PDA and was uniformly distributed along the MFC. The thermal conductivity of the pBN-MFC-PVA composite film increased with the addition of a pBN-MFC novel powder. MFC acted as “guides” to mitigate the h-BN agglomerate. In addition to the possible usage in the pBN-MFC-PVA composite film itself, the pBN-MFC hybrid powder may be a potential filler candidate for manufacturing thermal interface materials and wearable devices or protective materials.

## 1. Introduction

Boron nitride (BN) can be classified into four categories according to geometric structures: cubic BN, hexagonal BN, rhombohedral BN, and wurtzite BN. Hexagonal boron nitride (h-BN) is a two-dimensional material that has attracted the attention of researchers, and in which the lattice structure closely approximates the structure of graphite (it is also known as white graphite) [1]. Among its advantages, h-BN offers chemical and thermal stability, high thermal conductivity, and good dielectric properties. Thus, it has been widely used for the modification of resinous materials, and especially for the preparation of thermally conductive materials [2,3]. However, due to poor compatibility between h-BN and the polymer matrix, it is difficult to soak h-BN in resin, which results in poor dispersibility of h-BN within the matrix. Furthermore, it is difficult to make a composite with high thermal conductivity, which would affect the formation of heat conduction pathways and increases thermal resistance. To address this issue, researchers have previously focused on the surface modification and orientation of BN [4,5,6].

A coupling agent [7,8] is commonly used to modify the surface of h-BN to improve its compatibility with the resin matrix. This is a complex process [9,10,11] that has been used to give h-BN a certain directivity, which improves the thermal conductivity of the material. Currently, the goal is to apply more environmentally-friendly surface treatment agents and develop easier preparation methods. To that end, research has increasingly focused on the use of cellulose.

Cellulose is a degradable and highly abundant natural polymer. Different processing methods can be used to prepare cellulose products into different forms, including fiber [12], micro-fibrils, nano-fibrils [13], and cellulose hydrogel [14]. These substances retain the original structure and properties of cellulose. Moreover, its structure results in special chemical and physical properties such as a high modulus, specific surface area, strength, and transparency [13,14,15], which assist in increasing the thermal conductivity of materials. Xia et al. [12] combined sisal fiber cellulose with h-BN, and then filled the cellulose with epoxy resin to produce a thermally conductive composite material with thermal conductivity of 6.418 W·m^−1^·K^−1^. Zhu et al. [16] prepared a dielectric nanocomposite paper with layered h-BN wires by nano-fibrillated cellulose. It had high thermal conductivity along the BN paper surface. However, microfibril has rarely been considered. Furthermore, in most studies, BN was not very tightly bound to the crystal cellulose, and the obtained materials often lacked mechanical properties.

Cellulosic waste products are mostly burned to eliminate fungi and other parasites. The waste products are then composted or crammed to rot. To date, there has been a lack of adequate practical technology for the re-use of abandoned leaves, combined with a lack of knowledge (i.e., within the farming community) about their use as an abundant, inexpensive, commercially viable, biodegradable, and renewable agricultural residue. Making rational use of abandoned leaves is an important issue for scientists. Christos Didaskalou [17] extracted and isolated oleuropein, which is the most prominent biophenol, from abandoned olive leaves. Indu Tripathi et al. [18] prepared carbon-core dual-shell nanoparticles using agave nectar to remove pharmaceutical residues from hospital effluents. In addition, after harvesting, pineapple leaf waste remain problematic for farmers to deal with. As such, there is a great demand to identify other end uses [19,20,21,22]. Osumanu H. Ahmed et al. [23] extract Humic acid (HA) from pineapple leaves and the HA can serve as a foliar spray (supplement soil application K fertilizers), which is a source of K for freshwater fish and can be used as a soil conditioner. Leao A. L. et al. [24] demonstrated that the pineapple residue can represent an interesting source of raw material for industrial applications for the production of composite materials, special papers, chemical feed stocks (bromelin enzyme), and fabrics. Composite film material is a kind of widely used material in our daily life. How to prepare film materials with different functions and excellent properties is a hot topic in current research. Tobias Benselfelt et al. [25] consisted of cellulose nanofibrils (CNF) entangled with the algae polysaccharides alginate or carrageenan to prepare a kind of supramolecular double network film, which maintained its barrier properties at elevated relative humidity and the extensibility and ductility made possible through hygroplastic forming into three-dimensional shapes. Gergo Ignacz et al. [26] prepared robust, ion-stabilized membranes from polybenzimidazole (PBI) and its blends with polymer of intrinsic microporosity (PIM-1) were fabricated for organic solvent nanofiltration. Levente Cseri et al. [27] prepared an anion exchange membrane with a good mechanical property and high perm selectivity based on a graphene oxide (GO) and a polybenzimidazolium nanocomposite. JiDong Xu et al. [28] constructed Wood auto-hydrolysates (WAH)-based films by mixing WAH and chitosan (CS). The barrier composite films had great mechanical properties, good transmittance, and a low oxygen transfer rate. It would become attractive in the food packaging application. Poly (vinyl alcohol) (PVA), as a common raw material of film materials, has been used in many fields such as medicine, coating, and packaging. However, the development of applications of this polymer has been restricted by its lack of strength. Mechanical properties were often sacrificed when improving thermal conductivity. Our objective was to attempt to improve the mechanical properties as well as thermal conductivity of PVA by developing novel composites [29,30].

In this study, h-BN platelets were modified with polydopamine (PDA). Afterward, we used the binding properties of PDA to combine h-BN with pineapple leaf microfibril cellulose (MFC, prepared in the laboratory) to form a thermally conductive composite powder. The prepared composite powder was then used to prepare a PVA composite film with both good thermal conductivity and flexibility.

## 2. Materials and Methods 

### 2.1. Materials

h-BN (10 μm) was obtained from Qingzhou Fangyuan Boron Nitride Factory (Qingzhou, China) (≥99%). Dopamine hydrochloride (DA-HCl) (≥98%) and tris(hydroxymethyl)-aminomethane hydrochloride (Tris, ≥99%) were purchased from Macklin. Poly(vinyl alcohol) was purchased from Changchun Chemical Co., Ltd. (Changchun, China) (≥99%). The MFC was prepared in the laboratory.

### 2.2. Preparation of Hexagonal Boron Nitride and Microfibril Cellulose Hybrid Powder

Tris (0.605 g) was dissolved in 500 mL deionized water, and the pH of the solution was adjusted to 8.5 [31]. After the addition of h-BN (2.000 g) to the solution, stirring was continued and the solution was sonicated at 60 °C and 600 W for 3 h using an ultrasonic signal transmitter (NH-1000) obtained from the Shanghai Hanuo Instrument so that the h-BN sheets were stripped. DA-HCl (0.200 g) was then added to the mixture with continued stirring and sonication for 3 h to allow the dopamine to polymerize and fully interact with the h-BN surface. After the reaction was complete, the mixed solution was subjected to vacuum filtration and washed with deionized water (50 mL each time, 5 to 6 times) until the pH was neutral. The mixture was then dried at 80 °C for 24 h to form pBN (h-BN modified with PDA). Thereafter, pBN (0.700 g) and MFC (0.200 g) were mixed and dispersed in deionized water (300 mL). This solution was ultrasonicated at 60 °C and 600 W for 5 h with an ultrasonic pulse time of 2 s. The mixture was then filtered under vacuum and dried to form a pBN-MFC hybrid powder.

### 2.3. Preparation of pBN-MFC-PVA Composite Film

Different amounts of pBN-MFC hybrid powder were added and distributed uniformly in PVA solution. The mixed pBN-MFC-PVA dispersion slurries was dropped onto a 100 mm × 100 mm clean glass substrate and were de-aired in a vacuum chamber under 40 °C to prepare the composited films.

### 2.4. XPS Analysis

X-ray photoelectron spectroscopy (XPS) of pBN platelets was conducted using ESCALAB 250Xi spectrometer (Thermo Fisher Scientific, Waltham, MA, USA) equipped with an Al anode (Al-Kα 1486.6 eV). 

### 2.5. Thermogravimetric Analysis

Thermogravimetric analyses were carried out using TG/DTA thermal analyzer (Mettler Toledo, Zurich, Switzerland). The performed experiments ranged from 50 to 800 °C at a heating rate of 10 °C·min^−1^ under nitrogen atmosphere.

### 2.6. Morphological Analysis

The morphologies of MFC, h-BN, and pBN-MFC hybrid powder, and the dispersion of pBN-MFC hybrid powder in the PVA matrix were observed by using tungsten filament scanning electron microscopy (SEM, ZEISS, Oberkochen, Germany).

### 2.7. FT-IR Analysis

Fourier Transform-Infrared (FT-IR) spectra were recorded on a FT-IR spectrum (Perkin Elmer, Waltham, MA, USA). 100 instrument at room temperature. Before scanning, samples were dried in a drying cabinet at 80 °C for 4 h to remove moisture. The samples were scanned over a range from 400 cm^−1^ to 4000 cm^−1^.

### 2.8. Mechanical Analysis

The tensile strength and elongation at break of the pBN-MFC-PVA film was characterized using universal electronic tension tester (WDW-5D) (Sankun, Yangzhou, China) in tension film mode with an extension rate of 10 mm·min^−1^. The films were cut into samples measuring 75 mm × 10 mm, and the sample thicknesses were measured using a digital external micrometer (accurate to 0.001 mm) (Shengce Instrument Co., LTD, Wenzhou, China). The measurements were conducted in triplicate and average values were calculated.

### 2.9. Hydrophobicity Analysis

The static contact angle measurements of the PVA films and pBN-MFC-PVA films were carried out at 25 °C in an OCA 30 goniometer (Dataphysics, Stuttgart, Germany). Five readings were taken for each sample in different positions.

### 2.10. Thermal Conductivity Analysis

The thermal conductivity of the composite sheets was measured by a thermal conductivity coefficient analyzer (TC 3000, Xiatech Electronic Technology Co., Xi’an, China). 

## 3. Results and Discussion

### 3.1. Modification of h-BN with PDA

Figure 1 shows the route that was followed to modify the surface of h-BN with PDA. h-BN contains a six-membered ring composed of alternating B atoms and N atoms with π-π conjugation. The polymerization of dopamine involves nucleophilic reactions and intermolecular rearrangements in weakly basic environments. Abundant amine, hydroxyl groups, and aromatic rings were created, with PDA engaging in strong π-π stacking with the sp^2^-hybridized rings of BN as well as undergoing Van der Waals interactions with the surface of h-BN [32].

Figure 2a shows dispersions of h-BN and pBN in water. Given its inert surface, h-BN tends to agglomerate and is suspended above the water. In contrast, pBN is well dispersed and stabilized in water due to the hydrophilic phenolic hydroxyl and amino groups of PDA [33].

XPS was employed to determine the chemical composition on the surface of h-BN platelets. The results are shown in Figure 2b and listed in Table 1. The two main peaks on the broad spectrum of h-BN correspond to N 1s (397.1 eV) and B 1s (189.2 eV), while the peaks of O 1s (531.8 eV) and C 1s (284.7 eV) were comparatively weak. The percentages of N and B atoms were 47.76% and 46.65%, respectively, and the C atom content was only 4.99%. After modification, the contents of N and B atoms in pBN decreased to 38.13% and 40.27%, respectively, whereas the percentages of C and O atoms increased to 18.36% and 3.23%, respectively, because PDA contained more C and O atoms. The peaks in the XPS spectrum of pBN were curve-fitted, and the chemical bonds correspond to the peaks at 284.6 eV, 285.4 eV, and 288.3 eV, which were -CH-, C-N/C-OH, and C=O/COOH, respectively. The C-N and C=O groups belong to PDA. These results indicate that the PDA was successfully introduced onto the surfaces of h-BN. TG analysis was performed to estimate the amount of PDA coated onto the surface of the h-BN. As shown in Figure 2c, the pBN showed similar weight loss to that of DA-HCl. The rate of weight loss began to increase above 215 °C, and the final weight loss was 3.46%. This indicates that pBN is stable below 215 °C. The amount of PDA on pBN was estimated to be 2.77%.

### 3.2. Morphologies of h-BN, MFC, and pBN-MFC Hybrid Powder

The SEM image of unmodified h-BN is shown in Figure 3a. h-BN sheets are stacked and irregularly dispersed due to intermolecular forces. Figure 3b shows the SEM image of crystals of the self-prepared MFC, which illustrates its smooth surface. The length of the elongated crystals is 50 ± 10 μm. Many pBN crystals are tightly attached to the surface of the MFC in Figure 3c, which is a result of the full contact and interaction of pBN with MFC during the exposure to an ultrasound. We speculated that the phenolic hydroxyl and amino groups from PDA would help form hydrogen bonds with the hydroxyl groups on MFC to facilitate binding between pBN and MFC. This water-based method for combining pBN with MFC is environmentally-friendly and practical for the surface treatment and surface activation of h-BN. The FT-IR spectra of h-BN, MFC, and pBN-MFC hybrid powder are exhibited in Figure 3d. The most distinctive peaks, i.e., those at 1372 and 817 cm^−1^, are attributed to B-N stretching and B-N bending, respectively [34]. The characteristic peaks of MFC on the spectrum of pBN-MFC hybrid powder disappeared nearly completely, whereas the two characteristic peaks of pBN remained, which indicates that most of the characteristic peaks of pBN could be found in the pBN-MFC hybrid powder. This confirms that the components were well combined and successfully constitute the final composite [35]. This is consistent with the structure shown in Figure 3c.

### 3.3. Mechanical Property and Flexibility of pBN-MFC-PVA Film

We added increasing amounts of the pBN-MFC hybrid powder to PVA to prepare a series of pBN-MFC-PVA composite films by using a solution method, and investigated the effect on its mechanical properties. We conducted the tests to determine the tensile strength and elongation at the break of these films (Figure 4a,b). The results show that, as the pBN-MFC hybrid powder content increased, both the tensile strength and elongation at break first increased and then decreased. When the additive amount was 30 wt%, the tensile strength and breaking elongation of the pBN-MFC-PVA film increased by 2.4% and 12.5%, respectively, compared with pBN-PVA, and increased by 112.1% and 122.0%, respectively, compared with the film consisting of pure PVA. Clearly, MFC has a large impact on the mechanical properties of materials. Synergetic effects between the hydroxyl groups (-OH) and carboxyl groups (-COOH) of one-dimensional MFC [2] and the hydroxyl groups (-OH) and amino groups (-NH_2_) of pBN ensure that the interface between pBN-MFC hybrid powder and PVA is highly integrated and this is considered to enhance the mechanical properties (Figure 4c). However, when the amount of pBN-MFC hybrid powder exceeds 30 wt%, the mechanical strength of the composite films begins to decrease. We speculated that, increasing the mass fraction of the filler beyond 30 wt% caused the pBN-MFC hybrid powder to start agglomerating in the PVA, which increased microscopic defects. All of the mechanical properties of the composite materials improved compared with those of pure PVA. Thus, it was necessary to further explore the mechanism of action. The mechanical flexibility of the pBN-MFC-PVA film is demonstrated in Figure 4d.

### 3.4. Hydrophobicity and Thermal Conductivity of pBN-MFC-PVA Film

Hydrophobicity is another factor that is known to limit the application of PVA [29,30]. Therefore, we examined the contact angle by preparing eight samples with increasing amounts of pBN-MFC hybrid powder additive for comparison purposes. Images of two of the contact angle measurements are shown in Figure 5a (Sample No. 1) and Figure 5b (Sample No. 8). The contact angle of the pure PVA film was only 34 ± 1° because of the large number of highly polar hydroxyl groups. As shown in Table 2 and Figure 5c, the hydrophilicity of the pBN-MFC-PVA film decreased as the pBN-MFC hybrid powder content increased. When the amount of additive was 30 wt%, the contact angle reached 52 ± 0.6°, which is an increase of 49.8% (Figure 5b). A comparison of Sample No. 2 and No. 4 indicated that, in the presence of MFC, the contact angle of Sample No. 4 was greater than that of Sample No. 2 to which only pBN had been added. This can be understood by considering that h-BN is a non-polar substance that can enhance the hydrophobicity of the material while h-BN tends to be reunited in the substrate. Although the surface of MFC contains hydroxyl groups, most of its surface was coated by h-BN (Figure 3c). In other words, the MFC ensured a more even distribution of h-BN in the PVA film. Therefore, for the same amount of filler, the film with MFC showed stronger hydrophobicity. In the preparation of composite films, we cannot guarantee every position to be flat, which leads to the derivation of error of the contact angle and the error was within permission. This phenomenon could also be attributed to the microstructure or other factors, and remains to be further discussed.

Figure 5d shows the thermal conductivity versus the mass fraction of pBN-MFC hybrid powder. The thermal conductivity of the pure PVA film was poor at 0.17 W·m^−1^·K^−1^. An increase in the pBN-MFC hybrid powder content increased the thermal conductivity from 0.17 W·m^−1^·K^−1^ to 4.61 W·m^−1^·K^−1^, which is approximately 27 times higher than that of the pure PVA film and 68.9% more than that of pBN-PVA.

## 4. Conclusions

In this work, PDA was utilized to functionalize and improve the chemical activity of h-BN. Our approach helped h-BN to tightly attach to the surface of the MFC. We investigated ways to prepare pBN-MFC hybrid powder, using h-BN modified with PDA to form pBN, which was then combined with MFC through strong π-π and Van der Waals interactions to form the pBN-MFC hybrid powder, as confirmed by XPS, TGA, SEM, and FTIR. This composite was subsequently added to PVA, which served as the polymer matrix. The as-prepared pBN-MFC-PVA film was investigated to evaluate the properties of samples with different mass fractions of pBN-MFC hybrid powder.

The pBN-MFC hybrid powder enhance the tensile strength and breaking elongation of the pBN-MFC-PVA film by 2.4% and 12.5%, respectively, compared with the pBN-PVA film. The contact angle of pBN-MFC-PVA film reached 52 ± 0.6° at 30 wt% pBN-MFC hybrid powder loading, which was an increase of 49.8% compared with the pure PVA film. Significant thermal conductivity of 4.61 W·m^−1^·K^−1^ was obtained at 30 wt% pBN-MFC hybrid powder loading, which is 27 times higher than that of pure PVA (0.17 W·m^−1^·K^−1^)) and 68.9% more than that of pBN-PVA.

The pBN-MFC hybrid powder could enhance the thermal conductivity as well as hydrophobicity of PVA without sacrificing mechanical properties. This novel composite film can be utilized in the preparation of thermal interface materials and the materials for wearable devices. The present drawback of the experiment is that we cannot ensure all h-BN combine with MFC and the best proportion of both materials is not clear yet. Future plans include elucidation of the experimental conditions for preparing pBN and combining it with MFC, including mechanistic research.

## Figures and Tables

**Figure 1 materials-12-00104-f001:**
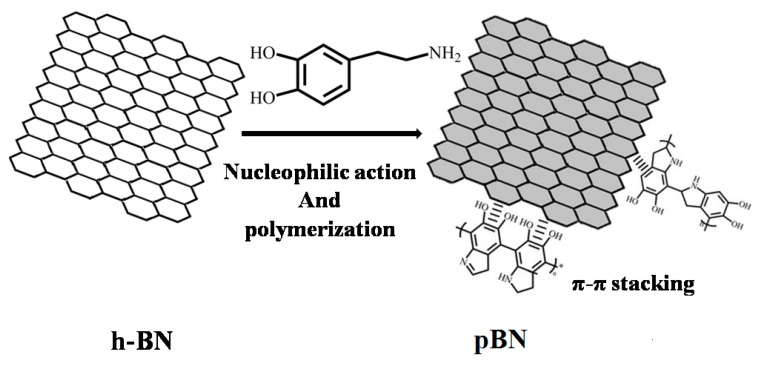
Schematic process of surface modification of h-BN with PDA.

**Figure 2 materials-12-00104-f002:**
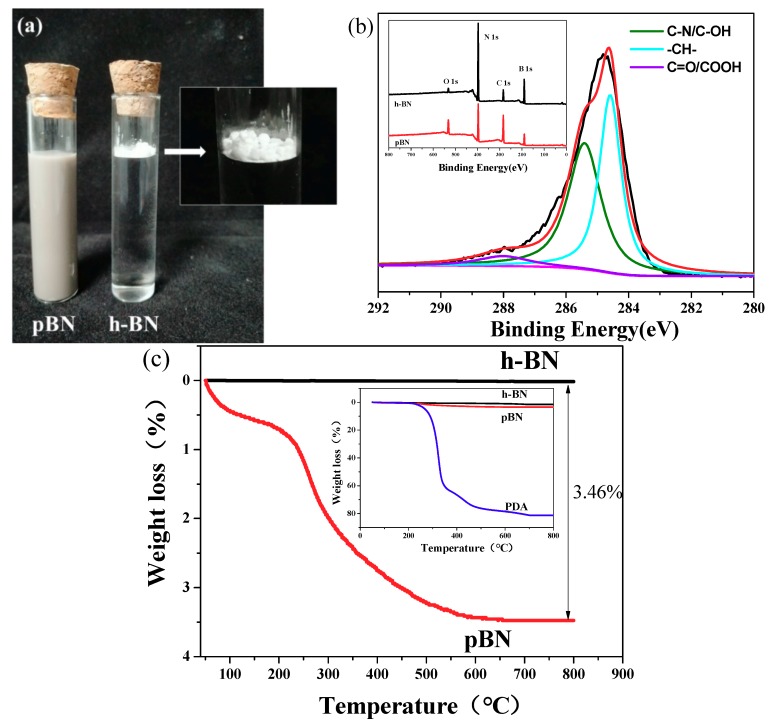
(**a**) Images of pBN (h-BN modified with PDA) and h-BN dispersed in water. (**b**) XPS survey scan for h-BN and pBN and C 1s high-resolution spectra for Pbn and (**c**) TGA curves of h-BN, PDA, and pBN.

**Figure 3 materials-12-00104-f003:**
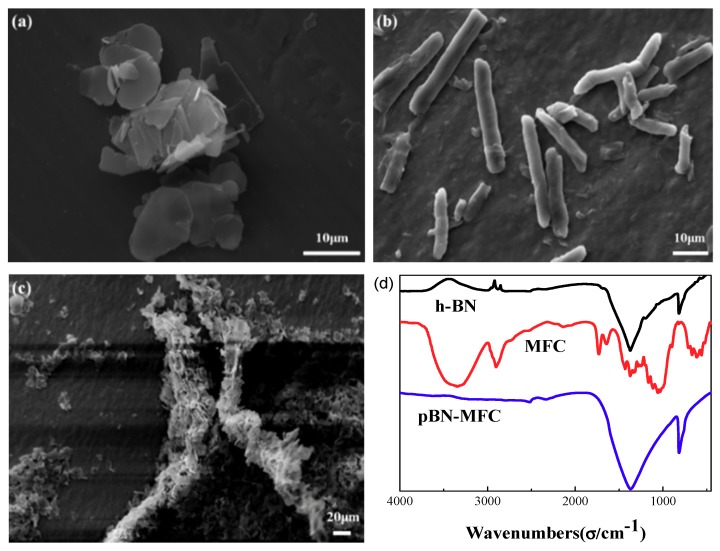
SEM images of (**a**) h-BN, (**b**) MFC, and (**c**) pBN-MFC hybrid powder. (**d**) FT-IR spectra of h-BN, MFC, and pBN-MFC hybrid powder.

**Figure 4 materials-12-00104-f004:**
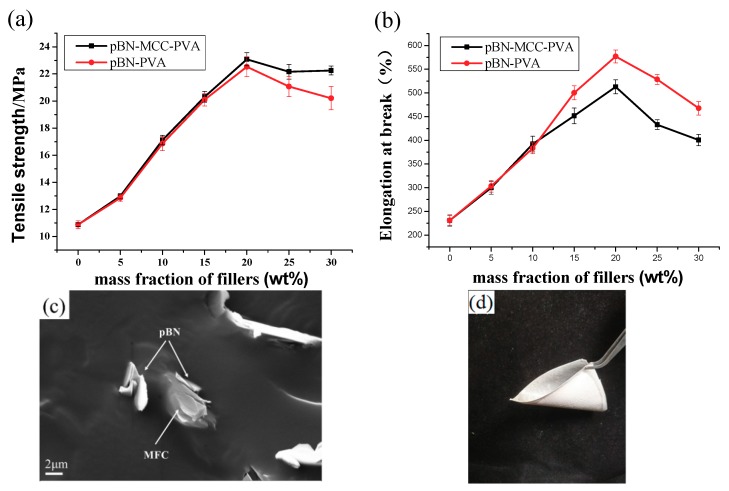
Mechanical properties of the (**a**) pBN-MFC-PVA and (**b**) pBN- PVA films. (**c**) SEM image of the fractured surface of the pBN-MFC-PVA film. (**d**) Image of the pBN-MFC-PVA film to show its flexibility.

**Figure 5 materials-12-00104-f005:**
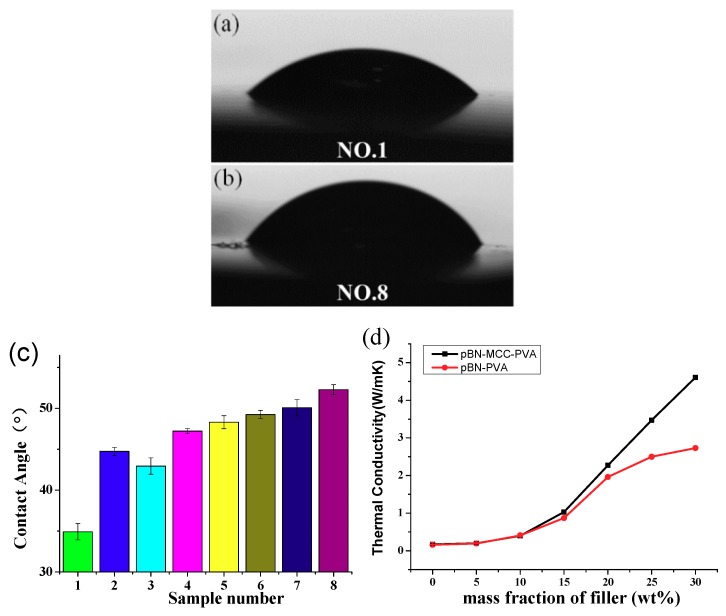
Contact angles of (**a**) pure PVA film (Sample No. 1), (**b**) pBN-MFC-PVA film (Sample No. 6), (**c**) Contact angles of the six samples including the pBN-MFC-PVA films. (**d**) Comparison of the thermal conductivity of the pBN-MFC-PVA films and pBN-PVA films.

**Table 1 materials-12-00104-t001:** Summary of XPS measurement results for h-BN and pBN.

Atomic Percentage (mol %)						
Sample	C 1s	O 1s	N 1s	B 1s	C/B	O/B
h-BN	5.0	0.6	47.8	46.6	0.10	0.01
pBN	18.4	3.2	38.1	40.3	0.45	0.08

**Table 2 materials-12-00104-t002:** Contact angle of each sample.

Sample Number	Sample Compositions	Contact Angles (°)	Standard Deviations
1	Pure PVA	34 ± 1	0.7
2	PVA + 10.0% pBN	44 ± 0.5	0.3
3	PVA + 5 wt% pBN-MFC hybrid powder	42 ± 1	0.8
4	PVA + 10 wt% pBN-MFC hybrid powder	47 ± 0.3	0.2
5	PVA + 15 wt% pBN-MFC hybrid powder	48 ± 0.8	0.5
6	PVA + 20 wt% pBN-MFC hybrid powder	49 ± 0.5	0.4
7	PVA + 25 wt% pBN-MFC hybrid powder	50 ± 0.7	0.5
8	PVA + 30 wt% pBN-MFC hybrid powder	52 ± 0.6	0.3

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
