# Peer review of "Hexagonal Boron Nitride/Microfibril Cellulose/Poly(vinyl alcohol) Ternary Composite Film with Thermal Conductivity and Flexibility"

_materials, 2018, doi:10.3390/ma12010104_

Reviewer 1 Report

The manuscript by Ge and co-workers discusses the preparation and characterization of composite films via waste utilization. The topic fits well in the scope of the journal and it is of interest to a broad audience. A reasonable amount of data is presented in the manuscript, and the results are interesting and novel. However, there are several major issues, which should be carefully addressed prior to further consideration for publishing the work.

1) Apart from the secondary interactions between BN and PDA, the authors claim covalent attachment of the PDA to the BN surface (scheme 1). However, there is no evidence presented in the manuscript that supports this hypothesis. The covalent attachment should be either omitted from the manuscript or direct evidence should be provided.

2) The reproducibility of the work should be demonstrated in the manuscript. Standard deviations should be reported for Table 1 and 2 as well as Figure 5c. The derivation of errors for all data should be clearly discussed in the text. Were independently prepared materials used to obtain the different datasets?

3) Avoid exaggeration, e.g. the “excellent” adjective in title should not be deleted.

4) Why is BN abbreviated but PVA not? It recommended not to abbreviate BN in the title to facilitate finding the article through search engines.

5) The purity/grade should be mentioned for all materials, chemicals and solvents used in the work, including BN and PVA. These can be mentioned under section 2.1 on Materials.

6) Line 85 should start as “2.2 Preparation of…”

7) The manuscript starts in medias res, and the authors should add a few sentences on composite films stressing their versatility and widespread application (nanopapers, DOI:10.1039/C8GC00590G; electrodialysis, DOI:10.1039/C8TA09160A; nanofiltration, DOI:10.1021/acsanm.8b01563; packaging, DOI:10.3390/ma11112264) before they narrow the scope own to BN/polymer films.

8) Pineapple-sourced products should be mentioned in the manuscript (DOI:10.1100/tsw.2004.199; DOI:10.1080/15421401003722930).

9) The agricultural waste utilization mentioned in line 70 are about two decades old, the authors should include some recent diverse examples (DOI:10.1039/C8TA05308A; DOI:10.1039/C7GC00912G).

10) The precise amount of washing solvent per mass of product should be included in the description of the methods under section 2.

11) The accuracy of the methods should be taken into account when reporting data. Significant digits should be considered, e.g. the % decimal places seem to be inappropriate.

12) Avoid using the ambiguous x/y formatting for units, and follow the IUPAC recommendation which is x y-1 throughout the manuscript, including figures and tables.

13) The conclusion section should include the main research findings in quantitative statements, and on the contrary the abstract should not have quantitative data showing the actual results.

14) Some critical comments about the proposed methodology should be mentioned at the end of the manuscript. What are the limitations and drawbacks of the methodology?

15) The reference list has some inconsistency in style, typos and incorrect names; proofread the entire list and follow the journal’s guideline on editing.

Author Response

Dear Reviewer,

Thank you for your comments. We have studied the valuable comments from you and tried our best to revise the manuscript which we hope meet with approval. Please download the revised version together with our responses to you. Welcome to offer valuable comments about our work again. Thank you.

Kind regards,

Mr. Ge

Reviewer 2 Report

This is an interesting paper on the combination of a waste product, a polymer and a ceramic.  Given the high cost of hexagonal boron nitride, it’s difficult to believe this would ever be an economic product.  This is just my editorial comment, and not a judgment on the quality of the research.

Correction of the English would greatly improve this manuscript. In certain passages, the authors’ meaning is difficult to discern, because the English is poor.

The very first sentence in the Introduction contains an error: there is no such material as “diamond BN.”  There is cubic boron nitride, which is similar to diamond, the all carbon material.  The one form the authors do not mention is rhombohedral boron nitride, which is a structural variant of hexagonal boron nitride.

In the third sentence of the Introduction, the word “thermal” is missing.  The sentence should read “As its advantages, h-BN offers excellent chemical and thermal stability, high thermal conductivity,…”  The word “conductivity” by itself implies electrical conductivity.   h-BN has very low electrical conductivity; it has a high electrical resistivity.

Section 3 on page 4

Start a new paragraph with the sentence beginning “XPS was employed….”

Section 3.1

Why is the word “remarkably” used in the sentence “After modification, ….remarkably, the percentages of C and O atoms…”  There isn’t anything obviously remarkable about this at all!

Figure 2 b

On the right hand side of this figure, there is a line with arrows on both ends labeled “1.86%,” but it is pointing to a distance that is approximately 3.5%.  This seems to be an error.  If it is not, then it needs to be better explained.

Section 3.3

The first five sentence in this paragraph describes background information on PVA.  Since these are not experimental results, this part should be moved to the Introduction. 

Author Response

Dear Reviewer,

Thank you for your comments. We have studied the valuable comments from you and tried our best to revise the manuscript which we hope meet with approval. Please download the revised version together with our responses to you. Welcome to offer valuable comments about our work again. Thank you.

Kind regards,

Mr. Ge

Round  2

Reviewer 1 Report

The authors have done a good revision but there are still major outstanding issues, which were all flagged during the previous submission:

Regarding previous comment #1, the authors refer to the π‐π stacking, which is correct. However, the scheme still shows covalent bonds between the PDA and the BN (top fight corner of the scheme, the benzene ring is linked with a covalent bond to the BN sheet). Since there is no evidence presented for any covalent bonding, the authors should refrain from such a depiction in the reaction scheme.

Regarding previous comment #2, the meaning of the presented errors are still not clear. The authors should include a statement in each caption explaining how the errors were derived, i.e. using independently prepared samples or the same batch was tested multiple time. Mention the number of measurements as well. This information is crucial to be clarified for the readers as it seems that the authors used the same samples, and the materials were not reproduced.

Regarding previous comment #10, the authors added "50 mL each time" but the number of times is not given. Therefore the total amount of liquid used is unknown. Include the number of times in order for the total volume to be disclosed.

Regarding previous comment #11, the authors should not rely on previous literature on determining the accuracy of the methods they used. The determination of accuracy of the methods employed are independent of other reports. The XPS atomic% should not be reported with two decimal places, even one decimal place is optimistic (in particular that there are no multiple measurements and errors reported).

Regarding previous comment #15, the list has still numerous errors, e.g. Ref 17 and 18: the surnames are abbreviated instead of the first names and spaces are missing after commas; Ref 19: the letter designation for affiliations were added as parts of the names of the authors; Ref 25 and 26: the surnames are abbreviated instead of the first names and a space is missing and "J." should be deleted; Ref 27: the surnames are abbreviated instead of the first names and one "J." should be deleted and the issue/pages should be added 6, 24728-24739; Ref 30: space and comma missing; Revise the entire list and correct all errors typos etc.

Author Response

 Dear Reviewer,

Thank you for your 2nd comments. 

We have studied the valuable comments from you and tried to revise the manuscript according to your reminding which we hope meet with approval. Please download the revised together with our responses to you. Welcome to offer valuable comments about our work again. Thank you very much.

Kind regards,
Mr. Ge

Round  3

Reviewer 1 Report

The minor corrections were done, however some spaces are still missing. This can be addressed during the proofreading or editorial typesetting.